# Integrated Gradients for Feature Assessment in Point Cloud-Based Data Sets

**Markus Schwegler** [1,*] [ID], **Christoph Müller** [1,2] and **Alexander Reiterer** [1,3] [ID]

1   Fraunhofer Institute for Physical Measurement Techniques IPM, 79110 Freiburg, Germany
2   Faculty of Digital Media, Furtwangen University, 78120 Furtwangen, Germany
3   Department of Sustainable Systems Enginneering INATECH, Albert Ludwigs University Freiburg, 79110 Freiburg, Germany
*   Correspondence: markus.schwegler@ipm.fraunhofer.de

**Abstract:** Integrated gradients is an explainable AI technique that aims to explain the relationship between a model's predictions in terms of its features. Adapting this technique to point clouds and semantic segmentation models allows a class-wise attribution of the predictions with respect to the input features. This allows better insight into how a model reached a prediction. Furthermore, it allows a quantitative analysis of how much each feature contributes to a prediction. To obtain these attributions, a baseline with high entropy is generated and interpolated with the point cloud to be visualized. These interpolated point clouds are then run through the network and their gradients are collected. By observing the change in gradients during each iteration an attribution can be found for each input feature. These can then be projected back onto the original point cloud and compared to the predictions and input point cloud. These attributions are generated using RandLA-Net due to it being an efficient semantic segmentation model that uses comparatively few parameters, therefore keeping the number of gradients that must be stored at a reasonable level. The attribution was run on the public Semantic3D dataset and the SVGEO large-scale urban dataset.

**Keywords:** point cloud; neural network; deep learning; integrated gradients; attributions; sensor fusion

## 1. Introduction

With the proliferation of point clouds for processing 3D data and the falling cost of Light Detection and Ranging (LiDAR), cameras, and other sensors that can generate point clouds, it becomes advantageous to combine multiple sensor types to increase the information acquired [1,2]. This could even consist of synthetically generated data not based on LiDARs but using existing geospatial data sources [3], thus vastly increasing the number of possible input features. Combined with large neural networks based on different approaches [4–7], it is difficult to evaluate which model fits best or what parameters to choose. This can either be performed manually by training each model with each combination of features, or on the other side of the spectrum using automated machine learning [8]. However, due to the nature of point clouds, the computational resources for both these methods are enormous. Therefore, we propose using the gradients generated while training a model to generate a quantitative evaluation of the impact each feature has on training. This has the added benefit of enabling some insights into how a network works and enable streamlining some hurdles faced when applying or refining machine learning models.

Significant progress has been made in this area when using data sets composed of sequences of discrete symbols, such as words or subword units used in natural language processing [9]. This allows for handling out-of-vocabulary words and capturing morphological information. Furthermore, it allows the network designer to learn whether a phrase has a positive or negative connotation such as the words "good" or "bad". Transferring

this sequential approach that relates to nuances and structures of the human language to the unstructured nature of point clouds and their geometric and spatial properties presents unique challenges and is one of the reasons why there has not been as much progress in this area.

We adapted the integrated gradients algorithm to be run on a large-scale urban point cloud dataset in order to gain greater insight into how a prediction is made and what influence the input features have on the prediction. RandLA-Net [10] was chosen as the network architecture to implement as it has 1/12th of the parameters compared to KPConv [6] and a similar mean Intersection over Union (mIoU) score. The low number of parameters is essential as the gradients accumulated by each pass scale with the number of parameters. Processing the gradients was implemented for two datasets composed of outdoor scenes, the first being Semantic3D, which is generated using a laser scanner. Moreover, the second is referred to as SVGEO as it was generated photogrammetrically by svGeosolutions GmbH using orthographic pictures taken from a drone and then annotated by Fraunhofer IPM.

## 2. Theory

### 2.1. Point Clouds

Recent advances in 3D imaging sensors, deep learning, and semantic segmentation have allowed the generation and processing of high-resolution 3D scans of the real world. These can range from small objects with extremely high resolution in the nanometre range to entire cities with centimeter accuracy. In a conventional image, all of the pixels contain information due to the nature of 2D data. Conversely, point clouds are very **sparse** as most of the volume in a 3D depiction is empty. Therefore, saving a mostly empty volume in a discrete form where each voxel has to be filled is very inefficient and not feasible for a large number of data points. However, by leveraging this, **unstructured** but more flexible format operations that require structure, such as finding neighboring points, become more complex. This is exacerbated by their **unordered** nature, making them invariant to permutation, these properties are visualized in Figure 1. Point clouds are also **irregular**; for instance, a 30 × 30 m point cloud might have any number of points, whereas a voxel representation always has a fixed number of voxels that are known in advance [11]. Due to these challenges, conventional 2D segmentation methods must be altered significantly to be successful when applied to point clouds.

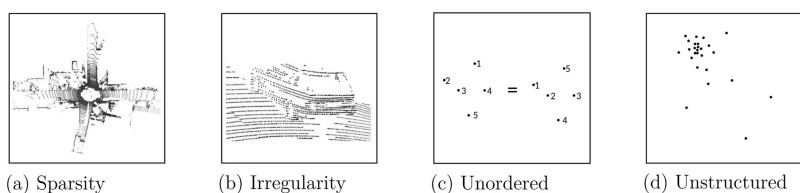

| (a) Sparsity | (b) Irregularity | (c) Unordered | (d) Unstructured |

**Figure 1.** Properties of point clouds visualized [12].

### 2.2. RandLA-Net

The RandLA-Net architecture is described as: "An efficient and lightweight neural architecture to directly infer per-point semantics for large-scale point clouds" [10]. Due to the Nature of large-scale point clouds containing millions of points and spanning hundreds of meters, those points must be progressively and efficiently downsampled in each neural layer. RandLA-Net achieves this by choosing the simplest and most efficient of all downsampling techniques; Random sampling. To mitigate the loss of useful point features a local feature aggregator is applied. As shown in Figure 2, the local feature aggregator consists of three neural units. In the context of this research, we define 'features' as the coordinates and Red, Green and Blue (RGB) values. However, it is important to note that the term 'features' is not limited to these parameters. Additional values, such as the

intensity recorded by the lidar, or other parameters gathered by accompanying sensors, could also be utilized along with the 3D coordinates to enrich the features.

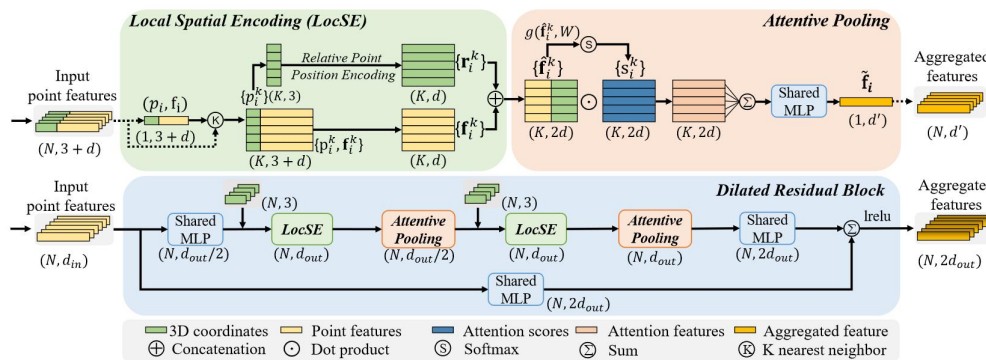

**Figure 2.** The local feature aggregator, as described in the RandLA-Net paper, in the top row the green and orange blocks show the concepts of the local spatial encoding and attentive pooling units, respectively. In the bottom panel, both units are chained together within a residual block [10].

### 2.2.1. Local Spatial Encoder

This unit explicitly encodes the coordinates of all neighboring points, this allows it to explicitly learn local geometric patterns. Operationally, this unit focuses on finding the nearest neighboring points using a k-nearest neighbors algorithm measuring Euclidean distances. These points are then fed into the relative point position encoding and explicitly encoded as follows:

$$r_i^k = MLP(p_i \oplus p_i^k \oplus (p_i - p_i^k) \oplus ||p_i - p_i^k||) \tag{1}$$

where $k$ is the number of points $\{p_i^1 \ldots p_i^k \ldots p_i^K\}$ nearest to center point $p_i$, $\oplus$ is the concatenation operation and $||.||$ the euclidean distance between the neighborhood end center points.

Encoding of the point features that correspond with the previously selected points is performed in a similar fashion. This results in a feature vector that is then concatenated with the point positions and results in an augmented feature vector $\hat{f}_i^k$. These explicitly learned geometric structures are then output by the local spatial encoder in the form of:

$$\hat{F}_i = \{\hat{f}_i^1 \ldots \hat{f}_i^k \ldots \hat{f}_i^K\} \tag{2}$$

### 2.2.2. Attentive Pooling

This neural unit aims to find attention scores that allow it to automatically select important features. The attention scores are generated using a shared MLP represented by the function $g()$ and $W$ being its weights as defined in the equation:

$$s_i^k = g(\hat{f}_i^k, W) \tag{3}$$

Formally the features are weighed as follows:

$$\bar{f}_i = \sum_{k+1}^{K} \hat{f}_i^k \cdot s_i^k \tag{4}$$

### 2.2.3. Dilated Residual Block

Increasing the receptive field of each point makes it more likely to preserve geometric details even as points are dropped during sub-sampling. Operationally, this is achieved by stacking multiple local spatial encoders and attentive pooling units and connecting them with skip connections, the resulting increase is visualized in Figure 3.

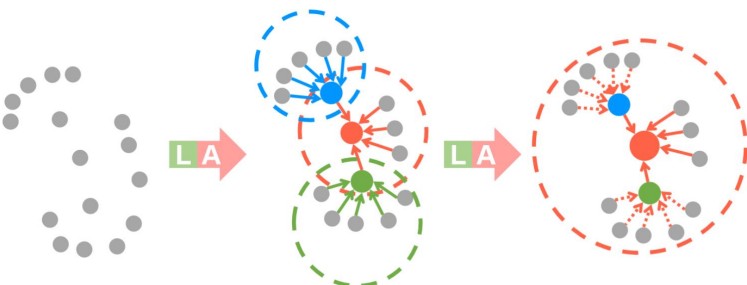

**Figure 3.** Demonstrates how the receptive field increases with each concatenation of the local spatial encoder and attentive pooling layers [10].

### 2.3. Sampling Point Clouds

Unlike 2D data, which are usually contained in the number of pixels present in a camera sensor, point clouds do not have a defined size. This leads to a problem when attempting to pass the data through a neural network, as the input has to have a fixed size or number of points depending on the architecture. For the RandLA-Net architecture, the number of input points must be fixed and the points have to be in a local neighborhood to conserve the geometric features. Therefore, generating small batches from the large point cloud is required. In this case, each batch contains the same number of points leading to batches as shown in Figure 4. These varying radii are a result of the underlying k-d tree being queried around the center point to return a fixed number of nearest neighbors. The center point is selected by a weighted sampling, here the weights are determined by decreasing them within the picked sphere in proportion to their proximity to the center point. These weights are updated after each selection. By setting a threshold for the largest weight, it can be ensured to continue sampling until each point was selected at least once.

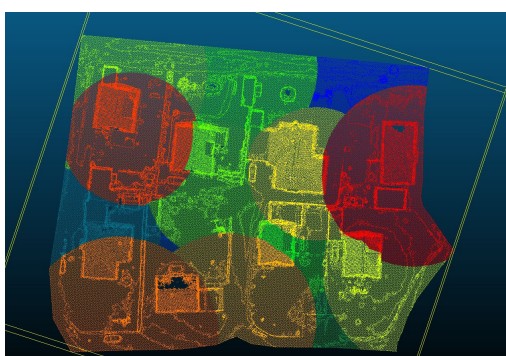

**Figure 4.** Different radii with a constant number of points.

Neighborhood

One of the largest performance constraints when dealing with point clouds is the fact that due to the lack of structure, finding the neighbors is not trivial and is usually performed in a KD-Tree [13]. This leads to a complexity of $O(NlogN)$, where N is the number of points in the point cloud. In contrast, when dealing with 2D data this is trivial as shown in Table 1 and its complexity does not scale with the number of pixels. To keep the visualization algorithm performant, it is essential to be able to look up the corresponding points with indexes. This can be achieved by allocating these indexes randomly, as the baseline was also randomly generated it does not violate any axioms of integrated gradients and allows very large point clouds to be processed. Another check to see if this is valid is whether the geometric structures predicted will line up with real geometric structures, this will be demonstrated later in the attributions generated.

**Table 1.** Indexing in a 2D image.

| $(i_{-1}, j_{-1})$ | $(i, j_{-1})$ | $(i_{+1}, j_{-1})$ |
|---|---|---|
| $(i_{-1}, j)$ | $(i, j)$ | $(i_{+1}, j)$ |
| $(i_{-1}, j_{+1})$ | $(i, j_{+1})$ | $(i_{+1}, j_{+1})$ |

### 2.4. Gradients

When training neural networks, the predictions that are output are compared to the desired outputs with the help of a loss function. This loss is then propagated through the entire network during the backward pass in the form of gradients and used to alter the network parameters along the way to better predict the data [14]. These gradients can be collected and, when compared to a baseline, provide insights into how the network learns. This method was proposed by axiomatic attribution for deep networks [15]. However, the method has to be modified to be applied to semantic segmentation [16], as the class-wise nature is essential for its functionality. Additionally, when using a point cloud, several alterations to the baseline have to be made.

The gradients generated by a network when predicting can offer valuable insights into its inner workings; however, visualizing them in a form that can be interpreted by humans is challenging, especially when dealing with three-dimensional data. This is due to the vast number of parameters used by deep neural networks, which cause a large number of gradients, combined with millions of points the volume of data becomes challenging to store and visualize. This is exacerbated by the fact that neural networks tend to transfer the input features into a higher, incomprehensible to humans dimension, and most of the prediction is performed in this feature space. Afterward, the network uses the same type of convolutions to get back to the three-dimensional input and allows the mapping of predictions to the original points. This does, however, transfer most of the inner workings of the network into a "black box", which cannot easily be analyzed. Obtaining the gradients used by the network, which determines how much of an influence each layer has on the input, and propagating them from the prediction backward to the output is currently the most promising approach to gaining some insight into this box.

### 2.5. Integrated Gradients

Two fundamental axioms of this method are sensitivity and implementation invariance, these are necessary to produce meaningful attributions. Sensitivity: "every input and baseline that differ in one feature but have different predictions then the differing feature should be given a non-zero attribution" [15] Implementation invariance: The network is treated as a black box, if input and output are the same so should the attributions.

Simply using the gradients by themselves without a baseline would not result in an accurate interpretation as the gradients can saturate during the prediction. Therefore, accumulating them across the path during predictions with interpolations between the baseline and the true point cloud allows a true representation of the influence of each feature on the prediction.

Baseline

A Baseline is motivated by its fundamental need in attribution as no discussion of cause is complete without a comparison [17]. This causation can be expressed in terms of counterfactuals, as described by Lewis in his theory of causation:

> "Where c and e are two distinct possible events, e causally depends on c if and only if, if c were to occur e would occur; and if c were not to occur e would not occur."

Therefore, if c were not to occur, e would not occur, in this case, c not occurring would be the change between the original image and the baseline. For a classification model, this baseline has to meet the requirements of having a high entropy across all classes.

In the Semantic3D dataset, we designate class zero as the "default class". Although this makes it challenging to understand why the model marked an object as 'unclassified', the broad definition of this class makes it difficult to gain any meaningful insights. This approach helps us avoid the issue of having to define a distribution that accurately represents the dataset, which could vary for each prediction due to class imbalances and significant differences in class distributions within point clouds. Although these differences can be minimized by sub-sampling point clouds during training, this approach is not practical during evaluation, as a contiguous volume is necessary for visualization.

An important characteristic of the chosen baseline is the number of steps (referred to as alpha) used in approximating the gradient. This value is influenced by the smoothness of the function being evaluated, the function being a result of the pixel gradients ad different interpolation steps. Given that each alpha requires a separate network pass and that the resulting gradients need to be stored for future processing, a small alpha value is preferred for the sake of performance and memory management. This is particularly important for 3D applications, which tend to have large memory footprints. The minimum feasible alpha value for both networks is determined experimentally, as shown in Figure 5.

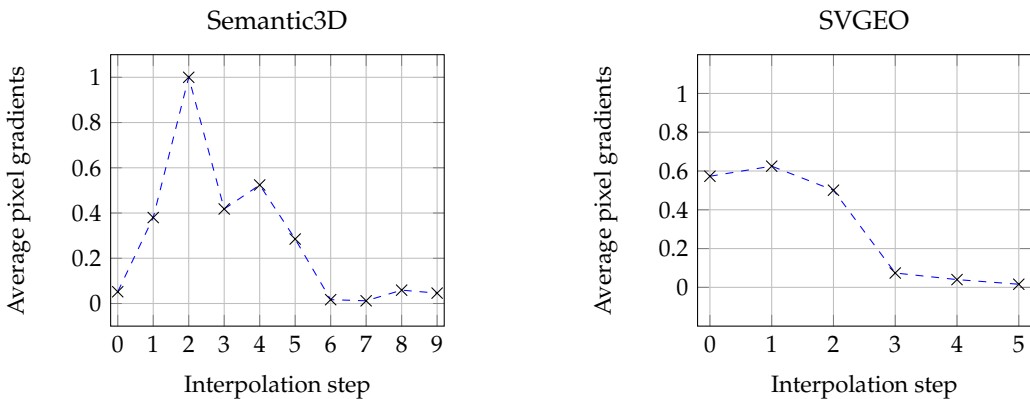

**Figure 5.** Convergence of average pixel gradients (normalized) for Semantic3D and SVGEO, the difference in interpolations is due to the size of the dataset making it more impossible.

Figure 5 demonstrates how the average normalized pixel gradients converge close to zero after seven iterations. The change in gradients is initially dramatic but quickly stabilizes. In our opinion, this atypically fast convergence is due to the vast number of points predicted in each interpolation, which smooths out many of the large differences found in typical applications for integrated gradients as normally only a couple of points are predicted in each network pass, i.e., for pictures or natural language processing. Rapid convergence is crucial when working with point clouds as even a small number of iterations can consume significant amounts of memory. For instance, the RandLA-Net algorithm, when performing 20 iterations on a point cloud of 5 million points, can require up to 100GB of Random Access Memory (RAM) during the attribution process. Therefore, using this technique with a large number of interpolations would not be practical.

### 2.6. Dataset SVGEO

The dataset mainly used in this paper was recorded by Fraunhofer IPM to improve the accuracy of rainwater simulations; therefore, most of it is outdoor and on a very large scale. The data were recorded using a drone flying over the area in a grid and taking vertical pictures. These overlapping images were then used to reconstruct a point cloud using Agisoft Metashape. These point clouds of entire cities and rural areas with a resolution of 3 cm were then subdivided and annotated. Due to the nature of the reconstruction, there are significantly more artifacts and errors than comparative LiDAR-based point clouds. For instance, sharp corners were somewhat smoothed and flat surfaces can have slight ridges making the detection of some small classes such as curbs very challenging.

As shown in Figure 6, vertical walls are almost always missing as they are not on any of the photos taken from a bird's eye view. However, when looking at the same section from the top, the point cloud looks almost flawless as seen in Figure 7.

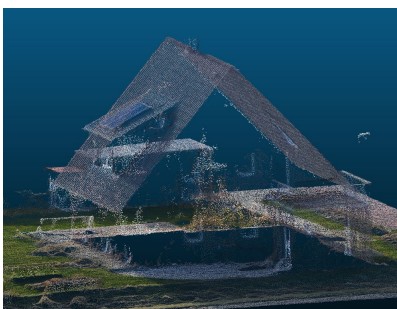

**Figure 6.** Sections of vertical point clouds missing.

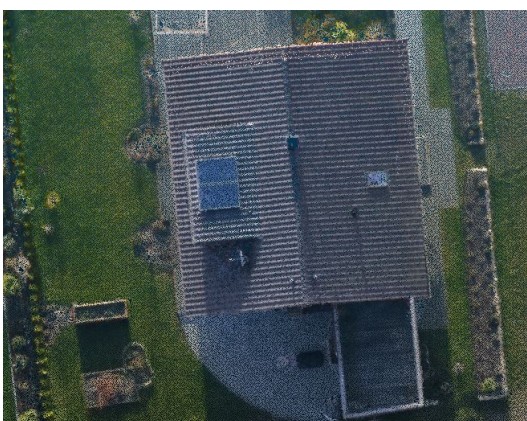

**Figure 7.** Bird's eye view of point cloud.

Figure 8 shows a typical artifact of point clouds generated using photogrammetry, the algorithm has difficulties matching points when lighting suddenly changes such as shadow edges, leading to substantial bumps in flat surfaces such as the roads as shown. This leads to incorrect geometries and causes confusion when relying on geometries as the model would have to gain an implicit understanding of this phenomenon to reach the correct prediction.

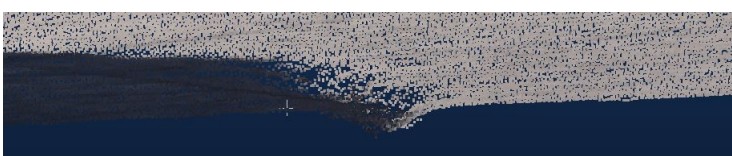

**Figure 8.** Artifact generated by photogrammetry.

## 3. Experiments

All experiments were conducted using the following hardware:

- CPU: INTEL CORE i7-6800K
- RAM: 128GB
- GPU type: NVIDIA GeForce RTX 1080ti

### 3.1. Training and Validation

The parameters used in training and gathering the gradients were adapted to our data set and GPU memory and are shown in Table 2:

**Table 2.** RandLA-Net training parameters.

| Parameter | Semantic3D | SVGEO |
|---|---|---|
| k nearest neighbors | 16 | 16 |
| number of points | 65,536 | 45,056 |
| number of classes | 8 | 19 |
| ignored label index | 0 | 0 |
| sub-sampling ratio | 4, 4, 4, 4, 2 | 4, 4, 4, 4 |
| input dimensions | 6 | 6 |
| feature dimensionality | 8 | 8 |
| output dimensions | 16, 64, 128, 256, 512 | 16, 64, 128, 256 |
| grid size | 0.05 | 0.06 |
| batch size | 2 | 3 |
| alphas | 20 | 20 |

The number of points chosen per iteration was adapted to the dataset to provide a large enough perceptive field, this is critical in allowing the network to learn structures. For both datasets, the aim was to have a receptive field that is large enough to fit entire buildings. Due to the nature of point clouds being unstructured, the radius varies with point cloud density. The distribution of these iterations was achieved by selecting center points from the point cloud until it was guaranteed that each point had been sampled at least once. This was performed by allocating weights to each point, which were increased when the point and its surrounding neighbors were selected until a bound was reached. These selected batches were then kept constant during the entire attribution.

*3.2. Integrating Gradients*

For each sampled batch a baseline was generated to create the interpolations necessary for the attribution.

Figure 9 demonstrates the resultant interpolated images for a subsection of the point cloud. The maximum values found in the original point cloud determine the first cuboid dimensions. Here, the same number of coordinates found in the original point cloud is randomly generated within these bounds, they are then allocated RGB values of 0,0,0 and form the baseline. The second point cloud shows the midway interpolation point between the baseline and the original point cloud; here, the coordinates roughly correspond to the house. However, a lot of noise can still be seen. In addition to the coordinates, the RGB values are also interpolated while the labels are kept as they must be constant when collecting the gradients.

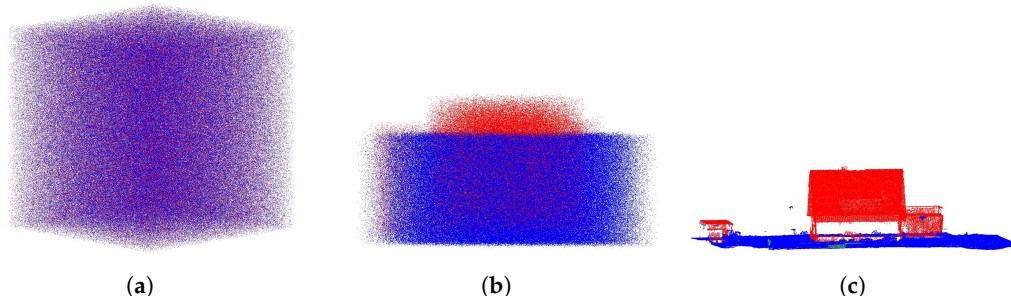

(**a**)                                            (**b**)                                            (**c**)

**Figure 9.** Three interpolation stages: baseline (**a**); midway point (**b**); and original point cloud (**c**).

To gather the gradients efficiently, the mini-batches had to be merged back into the original point cloud as many points were run multiple times. This was performed by

simply overwriting the point-wise gradients with the last collected gradient as all other methods proved too memory intensive.

Visualization

In order to visualize the result of the attributions created by integrated gradients, they were projected back into the original point cloud. To not clutter everything up, attributed features were added and allocated to each point. This results in a class-wise point cloud similar to the one generated by the prediction but with continuous values instead of discrete classes. The two most frequent and the least frequent class were chosen to display the differences in attributions per class, class 0 was ignored as it is too broad to give an accurate depiction.

Figure 10 shows what features in the point cloud the networks focus on. For all three classes, the roof of the houses is a significant landmark; however, the value of the attribution differs. As the network only truly differentiated between a building and not a building, the predictions can be treated as binary for this network and the middle class as another interpolation. Comparing "buildings" and "small wall opening with roof" shows a pattern, most strong gradients are inverted, this shows that the network learned to distinguish between building and not building by learning geometries in the point cloud, especially the shape of the gabled roof. Additionally, looking at the prediction in Figure 11, it becomes apparent that the two buildings on the right are predicted correctly, while the two buildings on the left are missing their walls in the prediction. It can be discerned that the gable has to have a strong positive attribution, while the walls need a strong negative attribution to be classified as a building. This gives humans some insight into why the network has made a prediction and allows parameters and networks to be refined while providing a greater understanding of the underlying model. A full flow diagram of the process running through the network is demonstrated in Figure 12

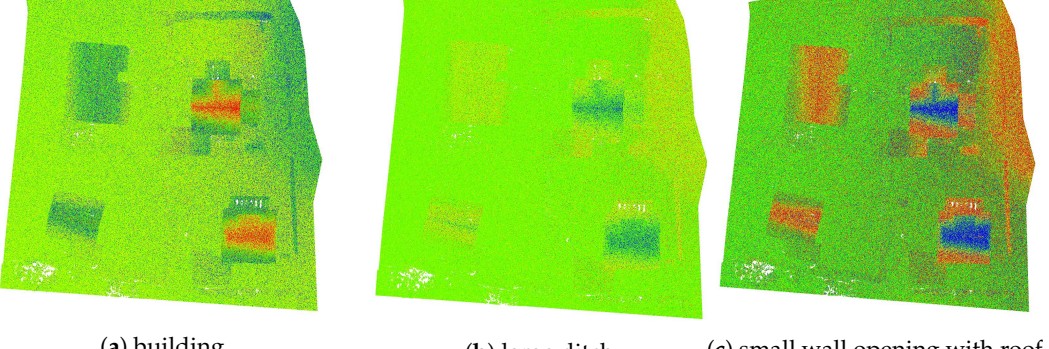

(**a**) building          (**b**) large ditch          (**c**) small wall opening with roof

**Figure 10.** point wise attributions by class, −10 is blue, 0 green, and 10 red, all other colors are interpolated between these three values.

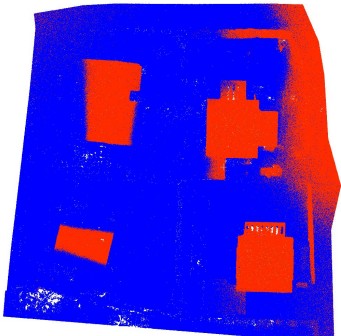

**Figure 11.** Predictions RandLA-Net without sub-sampling included to allow a comparison with the visualization of this prediction.

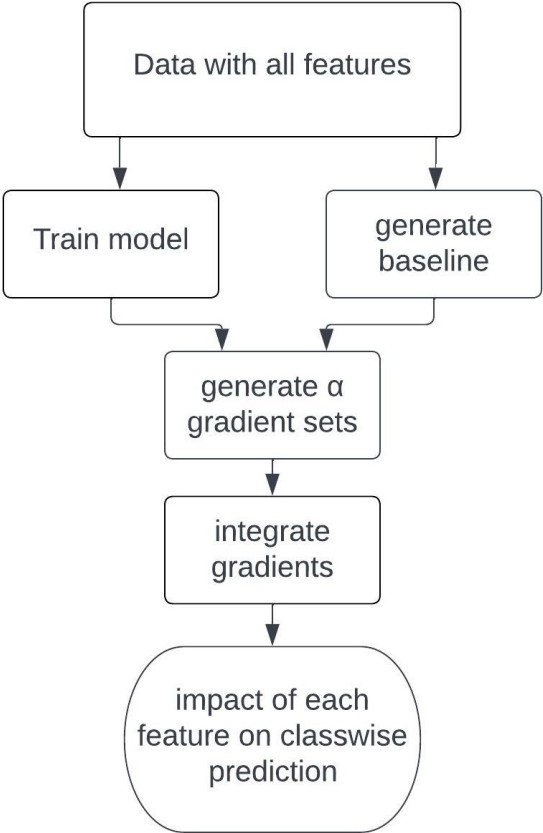

**Figure 12.** Flow chart for obtaining feature-wise attributions.

### 4. Impact of Each Modality on Final Prediction

As discussed in the previous section, integrated gradients were used to better visualize which features have the largest effect on the class-wise prediction. The network mainly trains on geometry, as demonstrated by Table 3, and the impact of the RGB color values had negligible influence on the predictions. This leads to the belief that using the current network architecture, RGB color values contribute relatively little to the prediction accuracy, they do however greatly increase the number of parameters used by the network, increasing complexity and computation time.

A meaningful comparison between the resulting attributions from training only the coordinates and then training the coordinates and RGB values could not be made as they are calculated relative to each other. However, existing literature supports the hypothesis that the RandLA-Net architecture primarily focuses on learning geometries. This has been demonstrated by running the network on the Semantic3D dataset. When the network was trained using only coordinates, the mIoU, was 77.71. In contrast, when RGB values were included in the dataset, the mIoU score only slightly increased to 81.03. When the network was trained on only RGB values, it failed to learn distinguishing features effectively. This test was run by setting all coordinates to random values, as completely removing the coordinates makes coherently sampling the point cloud impossible.

As displayed in Table 3, the attributions given to the coordinates vastly outweigh the RGB values. This is true for all classes; however, the differing attributions between the classes could also be interpreted as how certain the network is in predicting this class. The score might also be a measure of how unambiguous the features that represent the class are. This can be helpful when designing a class list as it allows systematic filtering out of ambiguous small classes, increasing the network's accuracy.

**Table 3.** Normalized and squared class-wise attributions for 10 iterations on the Semantic 3D dataset.

| Class | RGB | Coordinates |
|---|---|---|
| man-made terrain | 0 | 0.5498444 |
| natural terrain | $6.0026236 \times 10^{-6}$ | 0.9657749 |
| high vegetation | $9.214191 \times 10^{-6}$ | 0.91953284 |
| low vegetation | $9.287189 \times 10^{-6}$ | 0.8924127 |
| buildings | $4.39797 \times 10^{-6}$ | 0.63412595 |
| hard scape | $4.5726824 \times 10^{-6}$ | 0.6900027 |
| scanning artefacts | $1.1220334 \times 10^{-5}$ | 1 |
| cars | $6.3846205 \times 10^{-6}$ | 0.8025473 |

## 5. Conclusions

Integrated gradients can provide a deeper insight into why a model reaches a prediction; however, calculating this interpretation of attributions is computationally expensive and requires storing vast tensors to keep track of all the gradients. This limits the visualization to medium-point clouds or models with few parameters. However, the technique can be applied to quantitatively evaluate how much an input modality contributes to a prediction, this can be used to evaluate whether input modalities are useful for a prediction. New LiDAR sensors that provide additional data such as humidity or temperature have become available. Here, integrated gradients can significantly cut down training time as it enables all modalities to be fed into the model during training and after the attributions are found to provide a relative impact score for each modality. Otherwise, the network has to be run for every single modality separately. As the run time of the proposed integrated gradient model is less than 25% of the training time, this is time-efficient when evaluating at least three modalities. Additionally, the technique can be useful in evaluating which data fusions are beneficial, especially when a large amount of data sources are available.

Overall, applying integrated gradients on point clouds provided insights into the models' prediction as it allowed an understanding of why some of the buildings were predicted correctly and others were not. These insights enable the designers of these systems to reduce the memory size of the required point cloud, remove unnecessary sensors, and reduce the number of parameters required by the neural network to reach a prediction. Due to the large memory requirement, the application remains limited and further advances in the field of explainable AI are necessary to obtain a better understanding of how deep learning models operate.

**Author Contributions:** Conceptualization, M.S.; methodology, M.S.; software, M.S.; validation, M.S.; writing—original draft preparation, M.S.; supervision, C.M., A.R.; project administration, C.M., A.R.; funding acquisition, C.M., A.R. All authors have read and agreed to the published version of the manuscript.

**Funding:** This research received no external funding.

**Data Availability Statement:** No new data were created or analyzed in this study. Data sharing is not applicable to this article.

**Conflicts of Interest:** The authors declare no conflict of interest.

**Abbreviations**

The following abbreviations are used in this manuscript:

| | |
|---|---|
| LiDAR | Light Detection and Ranging |
| GPU | Graphics Processing Unit |
| CNN | Convolutional Neural Network |
| RGB | Red, Green, and Blue used to represent colors |
| IoU | Intersection over Union |
| MLP | Multi Layer Perceptron |
| IPM | Institute for Physical Measurement Techniques |

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
