# Peer review of "Integrated Gradients for Feature Assessment in Point Cloud-Based Data Sets"

_algorithms, doi:10.3390/a16070316_

Round 1

Reviewer 1 Report

In the paper, the authors use the integrated gradients technique to investigate the attribution of input features on the point cloud semantic segmentation results.  The authors applied the integrated gradients technique to one neural network architecture (RandLA-Net) and concluded that this architecture focuses mostly on 3D point coordinates and not on RGB values in the point cloud semantic segmentation class.

The main problem is that the description of the proposed approach is very difficult, and in many parts impossible, to understand. Authors write about evaluating the impact of 'features', but it's not explained in the body of the paper what these 'features' are. Do authors mean features like 3D  coordinates, intensity, and the number of LiDAR returns? Only at the end of the paper (Section 4) there's some clue that these 'features' are 3D point coordinates and RGB values.

The part of the Introduction talking about NLP is wrong. BERT model [8], was not trained on a limited data set, it was trained on a very large text corpus. And words in BERT are not one-hot encoded. BERT uses a subword tokenization scheme, and each token (=part of the word) is represented as an embedding.

Section "2.3. Sampling point clouds" does not give any detail on how the large input point cloud is sampled to create smaller point clouds with the same number of points.

Section 2.5.1 cannot be understood.

It's difficult to figure out what contributions are and what is the novelty of the paper. Applying a known technique (integrated gradients) to one neural network architecture isn't specifically novel. 

Furthermore, it would be good to confirm the conclusions using an alternative approach. E.g. compare the results obtained by training a network with 6-dimensional input (3D coords + RGB) versus 3-dimensional input (3D coords only). Do the results support the conclusion that RGB inputs are mostly ignored when using  RandLA-Net for point cloud semantic segmentation?

The English require extensive editing, as some sentences are impossible to understand, and many sound unnatural.

Some examples:

"In this case the batches each contain the same number of points leading to batches as shown...." rather it should be "each batch"

 "Two fundamental axioms of this method are sensitivity and implementation invariance, they enable the method to only display the network differences and not the difference in the attribution"  cannot be understood. I thought authors want to find attributions of each feature - but this sentence says that the method does not show the difference in the attribution.

The majority of the discussion in Section 2.5.1 cannot be understood. 

"The final characteristic of the baseline selected is the number of steps used in the gradient approximation".  What 'gradient approximation' do authors mean?

"This number referred to as alpha is determined by the smoothness of the underlying function." What "underlying function" do authors have in mind? Some words are unnecessarily written with a capital letter, e.g. "reconstruct a Point cloud"  

Author Response

Thank You for your feedback,

Point1: The main problem is that the description of the proposed approach is very difficult, and in many parts impossible, to understand. Authors write about evaluating the impact of 'features', but it's not explained in the body of the paper what these 'features' are. Do authors mean features like 3D  coordinates, intensity, and the number of LiDAR returns? Only at the end of the paper (Section 4) there's some clue that these 'features' are 3D point coordinates and RGB values.

The features are now explained when introducing the network architecture (line 74)

Point 2: The part of the Introduction talking about NLP is wrong. BERT model [8], was not trained on a limited data set, it was trained on a very large text corpus. And words in BERT are not one-hot encoded. BERT uses a subword tokenization scheme, and each token (=part of the word) is represented as an embedding.

The comparison between NLPs and 3D convolution network input spaces was reworked to clarify their differences. (line 31)

Point 3: Section "2.3. Sampling point clouds" does not give any detail on how the large input point cloud is sampled to create smaller point clouds with the same number of points.

The sampling was described in more detail (line 110)

Point 4: Section 2.5.1 cannot be understood.

The section has been revised to provide a more coherent and comprehensive understanding of the topic.

Point 5: It's difficult to figure out what contributions are and what is the novelty of the paper. Applying a known technique (integrated gradients) to one neural network architecture isn't specifically novel. 

The novelty is in its application of using integrated gradients not to understand how the network came to the conclusion but to use this technique to reduce the size of networks and input point clouds needed when processing data with any number of input features, provided only a small subset of these features is actually used in the prediction.

Point 6: Furthermore, it would be good to confirm the conclusions using an alternative approach. E.g. compare the results obtained by training a network with 6-dimensional input (3D coords + RGB) versus 3-dimensional input (3D coords only). Do the results support the conclusion that RGB inputs are mostly ignored when using  RandLA-Net for point cloud semantic segmentation?

As the attributions are relative within the network comparing two different networks, even just using different input parameters would not allow a meaningful comparison between the attributions. This section was extended to clarify this further (line 282)

Reviewer 2 Report

In this paper, a point cloud data algorithm named comprehensive gradient is proposed, which evaluates the quality and importance of each feature by observing the gradient changes during each iteration, and further explains the relationship between model features and prediction. The proposed method is novel, but there are some scientific questions the authors should consider further:

1): In the Introduction section, the existing researched need to be carefully summarized and elaborated. Currently, the introduction of relevant research and related research done by predecessors are less elaborated.

2) The results should notably exhibit the advancement of the proposed method with visualization result delivering and more clearly point cloud smoothing diagram presenting.

3) Fig 2 only manifests a fundamental structure similar to PointNet++, which not displays full related goals revolving around the theme of the paper, e.g., accurate gradient assessment or noise removal.

4) A work named “Individual Tree Crown Segmentation and Crown Width Extraction From a Heightmap Derived From Aerial Laser Scanning Data Using a Deep Learning Framework” affords a similar work with more detailed descriptions regarding the deep learning techniques processing point clouds, which maybe provide you some suggestions concerning a more effective way portraying diagrams and conveying information with a more clear manner.  

Author Response

Thank you for your feedback.

Point 1:  In the Introduction section, the existing research need to be carefully summarized and elaborated. Currently, the introduction of relevant research and related research done by predecessors are less elaborated.

Added more current research to the introduction section

Point 2: The results should notably exhibit the advancement of the proposed method with visualization result delivering and more clearly point cloud smoothing diagram presenting.

There are some visualizations in figure 10, getting larger images from the point clouds proved to be very cluttered.

Point 3: Fig 2 only manifests a fundamental structure similar to PointNet++, which not displays full related goals revolving around the theme of the paper, e.g., accurate gradient assessment or noise removal.

Figure two only provides a small snippet into the underlying RandLA-Net architecture.

Point 4: A work named “Individual Tree Crown Segmentation and Crown Width Extraction From a Heightmap Derived From Aerial Laser Scanning Data Using a Deep Learning Framework” affords a similar work with more detailed descriptions regarding the deep learning techniques processing point clouds, which maybe provide you some suggestions concerning a more effective way portraying diagrams and conveying information with a more clear manner. 

We had an in-depth look at this paper and used some inspiration from it for other types of applications for our proposed methodology. The synthetic generation of tree crowns sounds promising, we will refer to it in further research.

Reviewer 3 Report

Integrated gradients (IG) method has been applied to point clouds (PC) only few times, and here fst time with a random sampling based method (RandLA). Integrated gradients could help in directing downsampling, and explaining feature importance. 

The memory consumption and time needed per data window size would be of interest, but this is still quite an early stage for IG, and it is interesting enough to see how the method managed the chosen 8 features and the given window size. An interesting approach would have been in varying the feature set somehow.

The work is not easy to reproduce by this document only, but it is more likely that the research community is more interested about the idea itself and the demonstrated efficiency, and would pursue many modifications and variations. 

Detailed comments: Changes to do are minor ones and could be accomplished by the editors (if they want to avoid one minor iteration).

p.3, L76: change one p_i^k to p_i^1 !

Improve the Fig. 9 caption. Also, follow editorial advice on labeling details of images, for example using (a), (b), (c): 

Figure 9. Interpolation steps, baseline, midway point, and original point cloud --> Figure 9. Three interpolation stages: baseline (a), midway point (b), and original point cloud (c).

Author Response

Thank you for your feedback.

We fixed the erroneous equation and improved the captions  of Figure 9

The memory consumption and time needed per data window size would be of interest, but this is still quite an early stage for IG, and it is interesting enough to see how the method managed the chosen 8 features and the given window size. An interesting approach would have been in varying the feature set somehow.

That is a good follow up topic, currently there are very few datasets with the required number of features and the code would have to be updated to reduce the memory requirements to run it with more features.

Round 2

Reviewer 2 Report

As a whole, I feel that the revisions and additions to the manuscript add value to both its readability and scientific merit. The authors did a nice job of integrated gradient feature for point cloud assessment. Overall, the elaboration is sound as it pertains to point cloud analysis. The manuscript is well written, and the revisions made to the current draft make it better suited for publication.